# Bovine $F_1F_o$ ATP synthase monomers bend the lipid bilayer in 2D membrane crystals

Chimari Jiko[1†], Karen M Davies[2*†], Kyoko Shinzawa-Itoh[3], Kazutoshi Tani[4], Shintaro Maeda[3], Deryck J Mills[2], Tomitake Tsukihara[3,5], Yoshinori Fujiyoshi[4], Werner Kühlbrandt[2], Christoph Gerle[3,5*]

[1]Institute for Protein Research, Osaka University, Osaka, Japan; [2]Department of Structural Biology, Max Planck Institute of Biophysics, Frankfurt am Main, Germany; [3]Picobiology Institute, Department of Life Science, Graduate School of Life Science, University of Hyogo, Kamigori, Japan; [4]Cellular and Structural Physiology Institute, Nagoya University, Nagoya, Japan; [5]Core Research for Evolutional Science and Technology, Japan Science and Technology Agency, Kawaguchi, Japan

**Abstract** We have used a combination of electron cryo-tomography, subtomogram averaging, and electron crystallographic image processing to analyse the structure of intact bovine $F_1F_o$ ATP synthase in 2D membrane crystals. ATPase assays and mass spectrometry analysis of the 2D crystals confirmed that the enzyme complex was complete and active. The structure of the matrix-exposed region was determined at 24 Å resolution by subtomogram averaging and repositioned into the tomographic volume to reveal the crystal packing. $F_1F_o$ ATP synthase complexes are inclined by 16° relative to the crystal plane, resulting in a zigzag topology of the membrane and indicating that monomeric bovine heart $F_1F_o$ ATP synthase by itself is sufficient to deform lipid bilayers. This local membrane curvature is likely to be instrumental in the formation of ATP synthase dimers and dimer rows, and thus for the shaping of mitochondrial cristae.

*For correspondence: karen.davies@biophys.mpg.de (KMD); gerle.christoph@gmail.com (CG)

†These authors contributed equally to this work

## Introduction

The $F_1F_o$ ATP synthase is a membrane-embedded nano-machine and a member of the rotary ATPases (F-, V- and A-ATPases), which are found in energy-converting membranes of all eukaryotes, bacteria, and archaea (*Muench et al., 2011*). The enzyme catalyses the formation of ATP from ADP and inorganic phosphate ($P_i$) using the energy stored in a trans-membrane electrochemical gradient of protons or sodium ions (*Boyer, 1997*; *von Ballmoos et al., 2009*). The bovine heart enzyme has a molecular mass of approximately 600 kDa and consists of 17 different subunits ($\alpha_3$, $\beta_3$, $\gamma$, $\delta$, $\varepsilon$, a, b, $c_8$, d, e, f, g, A6L, $F_6$, oligomycin sensitivity conferral protein [OSCP], DAPIT, and a 6.8 kDa protein) (*Meyer et al., 2007*; *Runswick et al., 2013*). The enzyme can be subdivided into four essential functional parts: the catalytic part, $(\alpha\beta)_3$, which binds and converts ADP and $P_i$ to ATP; the membrane-embedded part (a, b, $c_8$, e, f, g, A6L, DAPIT, and the 6.8 kDa protein) through which protons move across the membrane; the central stalk, $\gamma\delta\varepsilon$, which transmits the rotation of the membrane-embedded rotor-ring ($c_8$) to the catalytic region; and the peripheral stalk (b, d, $F_6$ and OSCP) which holds the catalytic part stationary relative to the membrane region. The catalytic and central stalk regions form the $F_1$ subcomplex and the remainder, the $F_o$ subcomplex.

Structural studies of the intact $F_1F_o$ ATP synthase complex have been held back by the tendency of the enzyme to dissociate when extracted from the membrane. Nevertheless, a number of atomic models have been obtained by x-ray crystallography for various parts of the yeast and bovine mitochondrial $F_1F_o$ ATP synthase including the $F_1$ subcomplex (*Abrahams et al., 1994*), the $F_1/c$-ring subcomplex (*Stock et al., 1999*; *Watt et al., 2010*), the peripheral stalk subcomplex (*Dickson et al., 2006*), and the

**eLife digest** Cells use a molecule called adenosine triphosphate (or ATP for short) to power many processes that are vital for life. Animals, plants, and fungi primarily make their ATP in a specialised compartment called the mitochondrion, which is found inside their cells. The mitochondrion is often referred to as the powerhouse of cells as it captures and stores the energy that animals gain from eating food in the molecule ATP. Other enzymes in the cell break apart ATP to release the stored energy, which they use to power various cellular processes.

The interior architecture of the mitochondrion includes a highly folded inner membrane where electrical energy is transformed into chemical energy. The tight folding of the inner membrane is thought to make this process more efficient. An enzyme named ATP synthase performs the final steps of the energy transformation process by producing ATP (ATP synthase literally means 'ATP maker'). This enzyme sits in pairs along the edges of the inner membrane folds. This raises the question: does the ATP synthase cause the membrane to fold or does this enzyme just 'prefer' these folded edges (which are instead caused by something else inside the mitochondrion)?

To investigate this question, Jiko, Davies et al. extracted ATP synthase from the mitochondria of cow hearts and mixed them with modified fat molecules to form a '2D membrane crystal': a membrane containing an ordered pattern of enzymes. An electron microscope was used to generate a three-dimensional volume of the 2D membrane crystal via a process similar to a MRI or CAT scan that one might have in hospital. In the three-dimensional volume of the membrane crystal, Jiko, Davies et al. discovered that instead of being flat as expected, the membrane of the 2D membrane crystal was rippled and that this ripple was caused by the membrane-embedded part of the ATP synthase. The geometry of the ripple exactly matched half of the bend at the edge of the membrane folds in the mitochondrion. Therefore, Jiko, Davies et al. concluded that a pair of ATP synthases, as found in mitochondria, was responsible for defining the tight folds of the inner mitochondrial membrane.

$F_1$/peripheral stalk fragment (*Rees et al., 2009*). All these structures except the *c*-ring belong to the soluble region of the $F_1F_o$ ATP synthase. The membrane-embedded part of the mitochondrial $F_1F_o$ ATP synthase has two important functions. First, it allows protons to cross the membrane, thereby driving ATP synthesis; and second, it causes the dimerisation and oligomerisation of the enzyme in the membrane. The first structure of the membrane-embedded region of an $F_1F_o$ ATP synthase dimer was recently determined for the colourless green algae, *Polytomella sp.*, by single-particle cryo-EM (*Allegretti et al., 2015*). At 7 Å, the map shows that the conserved *a*-subunit forms a pair of horizontal helix haripins adjacent to the *c*-ring, whereas the peripheral stalk forms an intricate and unique dimerisation interface which is different from that of the bovine or yeast enzyme (*Davies et al., 2011*).

In mitochondria, $F_1F_o$ ATP synthases occur as rows of dimers on highly curved ridges in cristae membranes (*Strauss et al., 2008*; *Davies et al., 2011*, *2012*). Knock-out studies in yeast demonstrated the involvement of subunits e and g in dimer formation and cristae morphology, suggesting a crucial role of the $F_1F_o$ ATP synthase dimers in cristae formation (*Paumard et al., 2002*; *Davies et al., 2012*). Subtomogram averaging of the $F_1F_o$ ATP synthase dimers in situ revealed an angle of 86° between the monomers (*Davies et al., 2012*). Coarse-grained molecular dynamic simulations have indicated that the shape of the $F_1F_o$ ATP synthase dimer alone is sufficient to deform the lipid bilayer and drive the self-assembly of the $F_1F_o$ ATP synthase dimers into rows (*Davies et al., 2012*). However, a recent single-particle cryo-EM map of detergent solubilised, bovine $F_1F_o$ ATP synthase monomers found the detergent micelle around the $F_o$ domain was bent leading to the suggestion that the monomer alone could deform lipid bilayers (*Baker et al., 2012*).

Electron cryo-crystallography and electron cryo-tomography allow the structure of membrane-embedded proteins to be investigated in a lipid environment (*Fujiyoshi and Unwin, 2008*; *Davies and Daum, 2013*). To determine the structure of membrane-embedded bovine heart $F_1F_o$ ATP synthase, we generated 2D crystals of intact and active bovine heart $F_1F_o$ ATP synthase using the synthetic lipid 1,2-dimyristoyl-sn-glycero-3-phosphocholine (DMPC). By a combination of

subtomogram averaging and electron crystallography of tomographic slices, we determined the in situ structure and packing of the enzyme complex in 2D crystals. The oblique orientation of individual complexes in the membrane results in a zigzag arrangement of the lipid bilayer, which perfectly matches the observed bend in the detergent micelle of the isolated complex (*Baker et al., 2012*). Our results demonstrate that the transmembrane region of bovine mitochondrial $F_1F_o$ ATP synthase monomer is sufficient to bend the lipid bilayer. This membrane deformation is likely to be a prerequisite for the self-association of $F_1F_o$ ATP synthases into dimers and dimers into rows, as observed in mitochondrial cristae.

## Results

### Formation of stable 2D crystals

Monomeric mitochondrial $F_1F_o$ ATP synthase was purified from bovine heart muscle tissue by sucrose density gradient centrifugation and ion-exchange chromatography. For 2D crystallisation, fractions exhibiting high oligomycin-sensitive ATPase activity (>95%) and a high content of native lipids (>100 lipid molecules per $F_1F_o$ ATP synthase) were mixed with synthetic DMPC. When the detergent was removed by dialysis, abundant crystalline vesicles formed that were stable for weeks (*Figure 1—figure supplement 1*). Analysis by SDS-PAGE and mass spectrometry confirmed that the crystalline vesicles contained all subunits of the bovine $F_1F_o$ ATP synthase, including the small $F_o$ subunits *e*, *f*, *g*, A6L, DAPIT, and the 6.8-kDa protein (*Figure 1—figure supplement 2*). ATPase assays and blue-native PAGE performed on digitonin-solubilised crystalline vesicles demonstrated that the $F_1F_o$ ATP synthase complex in the 2D crystals was intact, active, and highly sensitive to oligomycin (>95%) (*Figure 1—figure supplement 3*).

### Subtomogram average of the $F_1F_o$ ATP synthase

The 2D crystals were insufficiently ordered for electron crystallographic processing despite numerous attempts to improve their order and size. Thus, to gain insight into the structure of the membrane-embedded $F_1F_o$ ATP synthase, we performed electron cryo-tomography and sub-tomogram averaging. In the tomographic volumes, the $F_1$ subcomplex appears as a 10-nm spherical density attached to the membrane by a thin stalk and is arranged in an alternating up-and-down orientation relative to the central lipid bilayer (*Figure 1A,B*). Tomographic cross-sections of crystalline vesicles embedded in thick vitreous ice (>200 nm) showed a zigzag morphology of the membrane (*Figure 2*). Fourier transforms of flat areas of crystalline vesicles showed that the degree of crystalline order varied over short distances. The most highly ordered regions were found in vesicles with a rectangular appearance, in which two membranes were closely apposed (*Figure 1C,D*). These ordered membrane regions were chosen for further analysis.

A total of 2100 $F_1F_o$ ATP synthase particles from the flattest regions of rectangular crystalline vesicles were manually selected and averaged by gold standard procedures (*Scheres and Chen, 2012*). According to the 0.5 FSC criterion, the resulting $F_1F_o$ ATP synthase average had a resolution of 24 Å (*Figure 3* and *Figure 3—figure supplement 1*). Both the peripheral and central stalks as well as the individual α and β subunits are clearly visible. The α and β subunits form a non-symmetrical hexamer around the central stalk with alternating long and short sides, allowing an accurate assignment to their respective density regions (*Figure 3C,D*, *Figure 3—figure supplement 2*).

The peripheral stalk is the strongest feature in the sub-tomogram average and is easily visible at contour levels >5σ above the mean. As in other cryo-EM maps (*Rubinstein et al., 2003*; *Lau et al., 2008*; *Baker et al., 2012*; *Davies et al., 2012*), the peripheral stalk of the $F_1F_o$ ATP synthase is arranged along the non-catalytic α/β interface and is offset towards the α-subunit, with which it makes contact midway along the stalk (*Figure 3*). The main contact of the peripheral stalk with the $(αβ)_3$ assembly occurs at the top of the complex (*Figure 3A,D*). Two distinct densities are visible in this region: one above the α-subunit next to the peripheral stalk and the other above the remaining α-subunits (*Figure 3D*, $d_1$ and $d_2$, respectively). At lower contour levels, the two densities merge immediately above the β-subunit leaving the centre of the hexameric ring open.

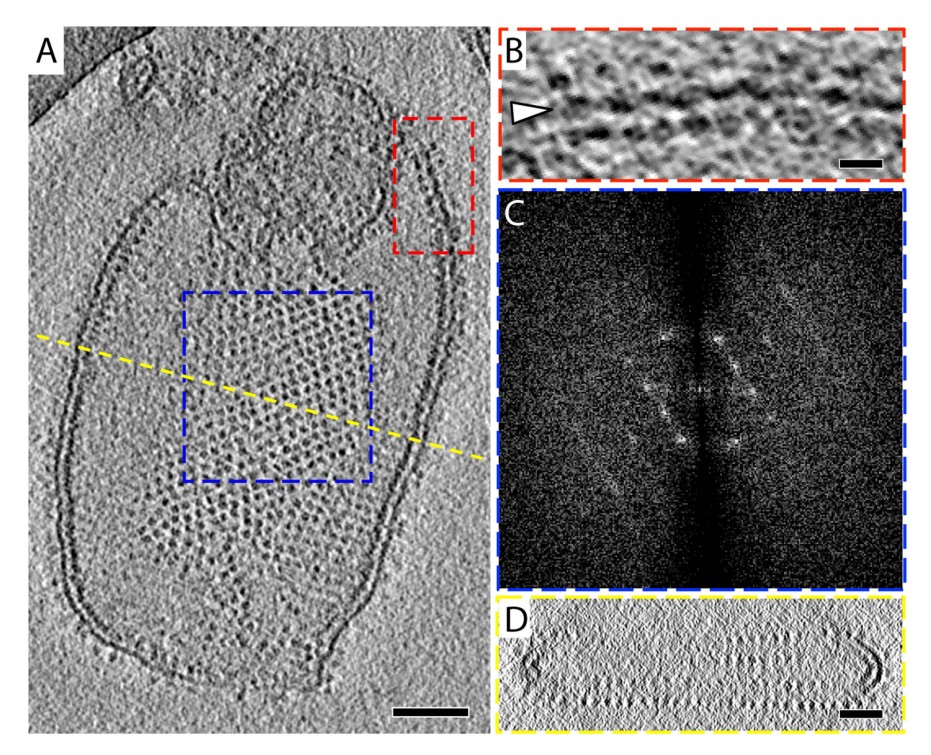

**Figure 1**. 2D crystals of $F_1F_o$ ATP synthase in vitreous ice. (**A**) Tomographic slice of a vesicle reconstituted with $F_1F_o$ ATP synthase. The ATP synthases appear as 10 nm spherical densities located 15 nm above the membrane. (**B**) Enlarged view of red boxed area in **A** showing the zigzag membrane structure (arrowhead). (See also *Figure 2*.) (**C**) Fourier transform of the blue-boxed area in (**A**). (**D**) Cross-section along the yellow dashed line in (**A**). Scale bar: (**A**) 100 nm, (**B**) 20 nm, (**D**) 50 nm.

The following figure supplements are available for figure 1:

**Figure supplement 1**. Crystalline vesicles of $F_1F_o$ ATP synthase in negative stain.

**Figure supplement 2**. Subunit composition of $F_1F_o$ ATP synthase isolated from 2D crystals.

**Figure supplement 3**. Blue-native polyacrylamide gel and activity assay of $F_1F_o$ ATP synthase isolated from 2D crystals.

## Fitting of atomic models

The first complex we docked into the subtomogram average was the x-ray structure of the $F_1$/peripheral stalk fragment (PDB: 2WSS *Rees et al., 2009*). All parts of this structure fitted well, with the N-terminal domain of the OSCP occupying the density immediately above the αβα subunits, and the C-terminal domain plus the peripheral stalk occupying the density above the remaining α-subunit. The peripheral stalk subcomplex (PDB: 2CLY *Dickson et al., 2006*) was then added to the fitted 2WSS structure but the resulting structure did not fit the map (*Figure 4—figure supplement 1*). The *b*-subunit residues 182–207 and the C-terminal domain of the OSCP subunit (residues 115–188) from the 2WSS model were then added to the x-ray structure of the bovine peripheral stalk subcomplex (PDB: 2CLY *Dickson et al., 2006*) and fitted as a rigid body into the subtomogram average. The new model resulted in an excellent match with the EM density (*Figure 4—figure supplement 1*). Finally the *c*-ring from the $F_1/c_8$ x-ray structure (PDB: 2XND *Watt et al., 2010*) was fitted by superimposing the $F_1$ subcomplex of this structure with that of 2WSS (*Figure 4*).

Although the sub-tomogram average was calculated from protein reconstituted into membranes, the lipid bilayer itself was not resolved in the average. This is an effect of the missing wedge of

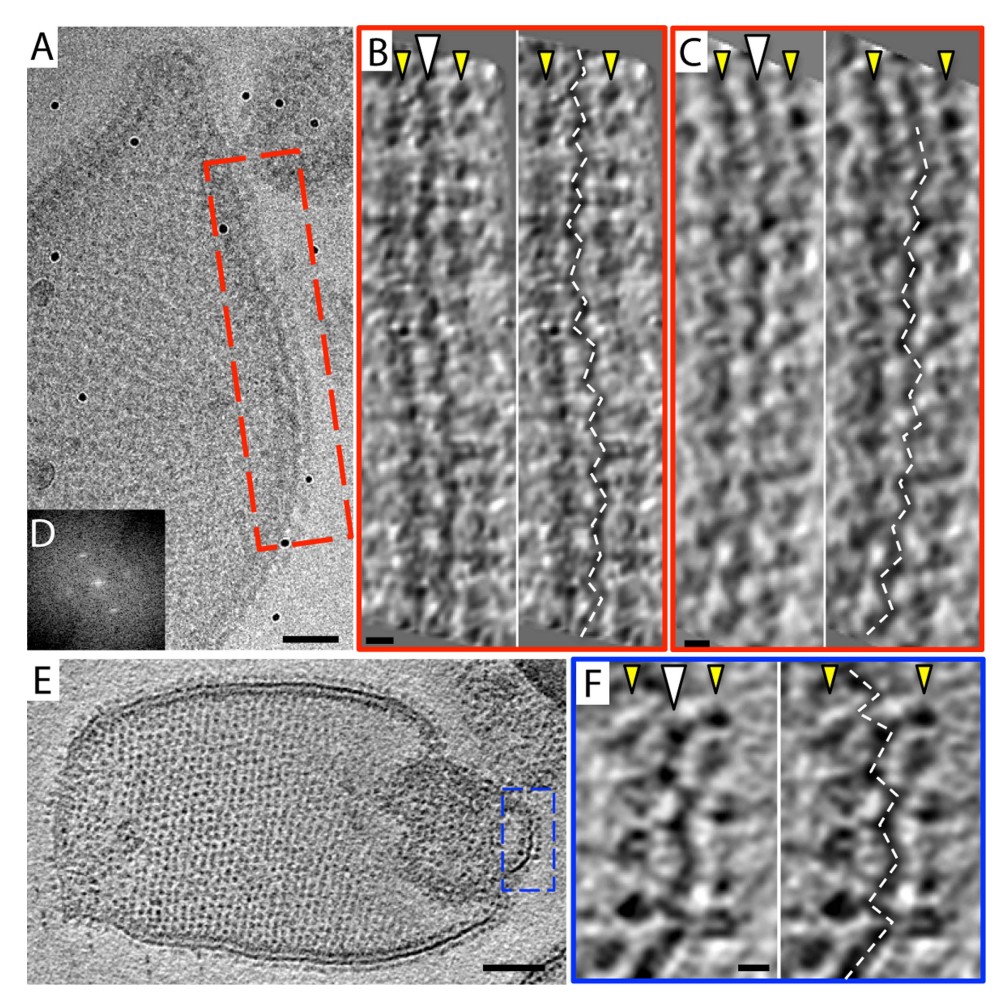

**Figure 2**. Zigzag membrane structure. (**A**) Projection image of a vesicle reconstituted with $F_1F_o$ ATP synthase. (**B**) and (**C**) tomographic slices of boxed area in (**A**) at different z-heights. (**D**) Fourier transform of an oblique tomographic slice through boxed area in (**A**) showing weak diffraction. (**E**) Tomographic slice of rectangular crystalline vesicle. (**F**) Close-up of boxed area in (**E**). White arrowhead, indicates membrane plane, yellow arrowhead, line of $F_1$ head groups. Scale bar: (**A**) 50 nm, (**B**, **C**, and **F**) 10 nm, (**E**) 100 nm.

information in the tomogram, which blurs out map features perpendicular to the electron beam, rendering the lipid bilayer in effect invisible (*Penczek and Frank, 2006*). An accurate estimate of the membrane position can however be obtained from the position of the *c*-ring in our fitted atomic model (*Figure 4*).

## Analysis of crystal packing

To assess the packing of $F_1F_o$ ATP synthases in the 2D crystals, the sub-tomogram average was re-inserted into the original tomograms using the inverse of the parameters calculated during averaging (*Pruggnaller et al., 2008*; *Video 1*). Defects in the 2D crystal lattice were apparent in the rotational orientation of individual complexes in a single layer (*Figure 5—figure supplement 1*). For a better understanding of the molecular packing, 400 particles were selected from a small region (235 × 285 nm) of a single crystalline layer, which showed sharp diffraction spots to the third order. These particles were averaged and refined against a reference model calculated from the selected particles, which had been masked to contain a 3 × 3 array of $F_1F_o$ ATP synthase densities. The box size of the final average was enlarged to include particles of opposite orientation in the same membrane and the

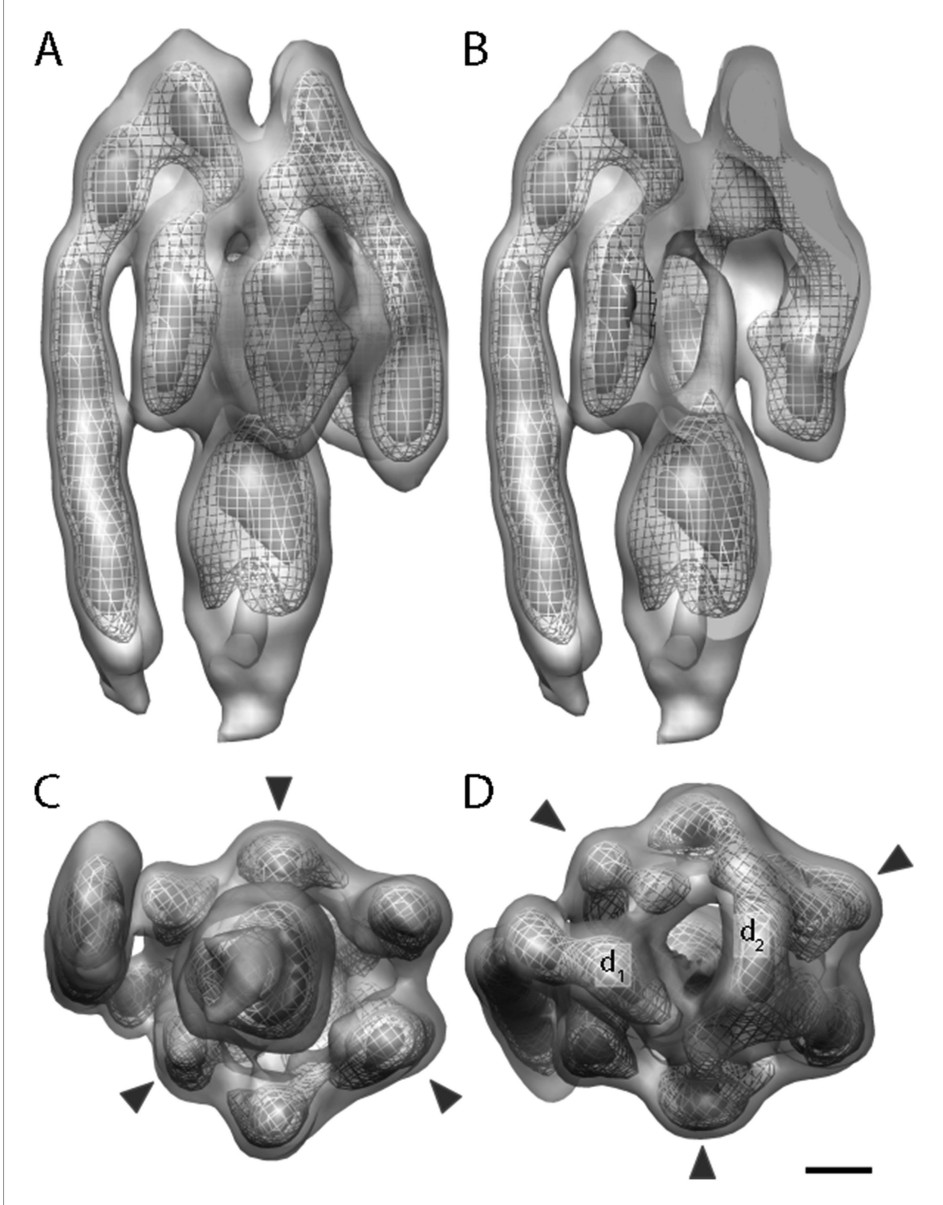

**Figure 3**. Sub-tomogram average of bovine heart mitochondrial $F_1F_o$ ATP synthase calculated from 2D crystals. (**A**) Surface view, (**B**) longitudinal section showing central stalk, (**C**) bottom view and (**D**) top view. Arrowheads: positions of the β subunits. $d_1$ and $d_2$: densities connecting peripheral stalk to $F_1$ subcomplex. Threshold levels: light grey, 1 σ; mesh, 3 σ; dark grey, 5 σ. Scale bar: 20 Å.
The following figure supplements are available for figure 3:

**Figure supplement 1**. Resolution estimate of the ATP synthase monomer sub-tomogram average.

**Figure supplement 2**. The α/β hexamer.

---

higher resolution subtomogram average (shown in *Figure 3*) was fitted multiple times into this volume (*Figure 5* and *Video 2*).

*Figure 5A* shows the typical packing of $F_1F_o$ ATP synthases in the most ordered regions of rectangular-shaped crystalline vesicles. The $F_1F_o$ ATP synthases form pairs of particles of twofold symmetry, which are in contact half-way up the peripheral stalks. The angle included by the long axes of the monomers in a pair is approximately 24˚ (*Figure 5B*). Note that these ATP synthase pairs in the

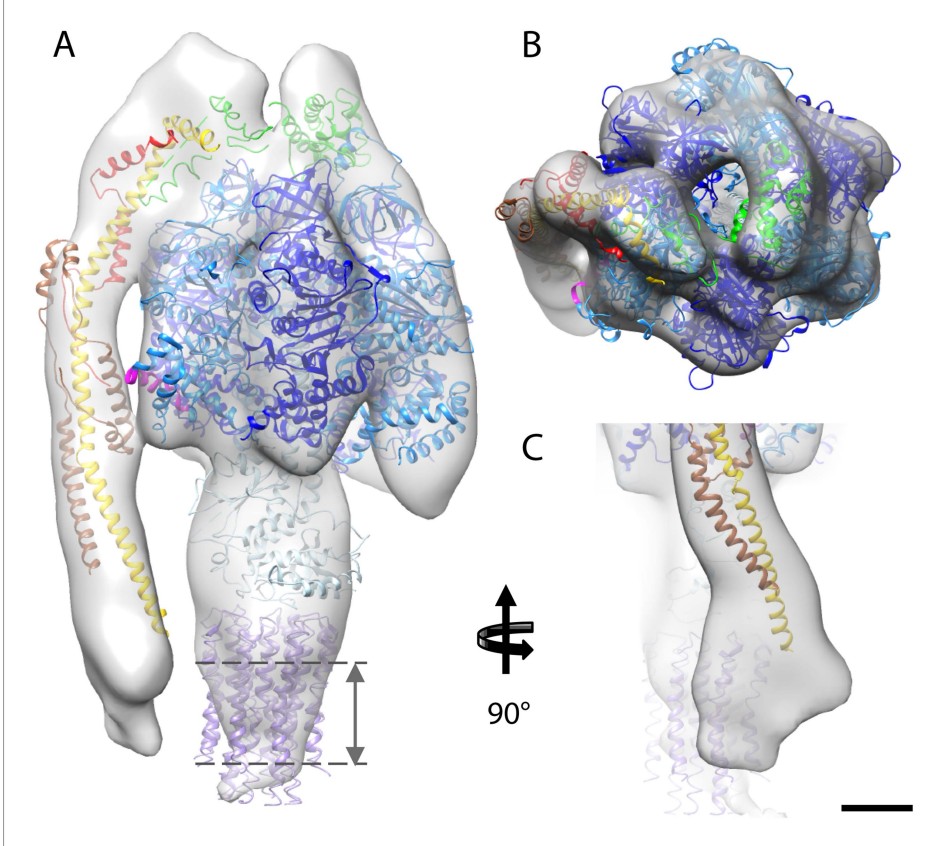

**Figure 4**. Fitted atomic model of the bovine $F_1F_o$ ATP synthase. (**A–C**) Sub-tomogram average with fitted atomic models (**A**) side view, (**B**) top view, and (**C**) the peripheral stalk. Blue, catalytic domain; grey, central stalk; green, oligomycin sensitivity conferral protein (OSCP) from PDB:2WSS (*Rees et al., 2009*). Purple, *c*-ring (PDB:2XND) (*Watt et al., 2010*); Yellow-red, peripheral stalk fragment (PDB: 2CLY) (*Dickson et al., 2006*) with additional residues from PDB:2WSS (*Rees et al., 2009*). Pink, α-subunit helix thought to interact with the peripheral stalk. Dashed lines, position of membrane. Scale bar, 20 Å.

The following figure supplement is available for figure 4:

**Figure supplement 1**. Possible atomic models fitted to sub-tomogram average.

2D crystals are structurally unrelated to the native dimers observed in mitochondrial membranes (*Strauss et al., 2008*; *Davies et al., 2011*). Pairs of $F_1F_o$ ATP synthase particles from opposite faces of the lipid bilayer interact via their *c*-rings and are related to each other by a ~90° rotation (*Figure 5A*).

The interaction of *c*-rings in the membrane is best observed when viewing a cross-section of the map along the crystallographic a-axis (*Figure 5C*). In this view, it becomes apparent that the pairs of opposing *c*-rings and the long axes of the $F_1F_o$ ATP synthases include an angle of 16° with the crystal plane. Therefore, in order for the *c*-rings to be fully embedded in the membrane, the lipid bilayer within the 2D crystals must adopt a zigzag topology, as observed in the vesicle cross-sections (*Figure 2*). To assess whether the $F_o$ domain of the bovine $F_1F_o$ ATP synthase could in fact account for this curvature, we placed the segmented volume of the single-particle map of this enzyme (*Baker et al., 2012*) into our extended sub-tomogram average. To our surprise, the bend of the micelle-embedded membrane region of the single-particle map perfectly matched the kink in the membrane that is required to hold successive *c*-rings together (*Figure 5D,E*). Thus the packing of $F_1F_o$ ATP synthases into the 2D crystal strongly supports the notion that the membrane domain of the monomeric bovine heart $F_1F_o$ ATP synthases is inherently bent and that this is sufficient to impose a local curvature on the lipid bilayer.

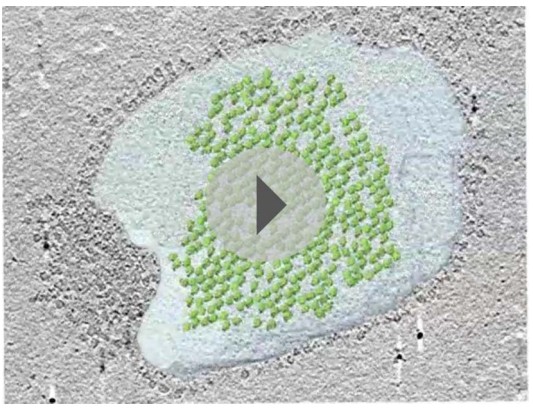

**Video 1.** Tomographic volume of a 2D crystal with re-inserted subtomogram average. The video moves through the z-stacks of the tomogram, revealing the location of the re-inserted sub-tomogram averages of the $F_1F_o$ ATP synthase. The four layers of $F_1F_o$ ATP synthase molecules are coloured in pink, blue, orange, and green. The predicted position of the vesicle membrane is shown in blue.

## Crystal packing analysed by crystallographic processing of selected z-slices

The subtomogram average lacks well-defined density in the membrane-embedded part of the complex. Therefore, we validated our packing model by crystallographic image processing. Z-slices through the tomographic volume at the levels of the catalytic $(\alpha\beta)_3$ hexamer, the stalk region and the membrane-embedded domain, which all showed sharp diffraction spots to the third order, were transformed into projection images and processed by electron crystallographic routines without applying symmetry (*Figure 6*, *Figure 6—figure supplement 1* and *Figure 6—figure supplement 2*) (*Henderson et al., 1986*; *Crowther et al., 1996*).

The resulting projection maps were plotted at 30 Å resolution and showed good correlation with the crystal-packing model (*Figure 6*). The catalytic $(\alpha\beta)_3$ hexamer map featured twofold symmetric particles that were connected via a small protrusion due to the peripheral stalk. The stalk region showed four distinct densities related by a twofold rotation. The larger, more distal density corresponded to the central stalk and the thinner, more elliptical density to the peripheral stalk. For the membrane region, where no clear protein density is observed in the subtomogram average, continuous density is observed in the direction of the proposed c-ring/$F_o$ interaction, but not in the perpendicular direction (*Figure 6*). Thus, the zigzag membrane geometry observed in *Figure 5C* was due to the membrane region of the $F_1F_o$ ATP synthase.

## Discussion

We have reconstituted intact bovine heart $F_1F_o$ ATP synthase into lipid bilayers. Due to the mild conditions during protein purification and reconstitution, the protein remained stable and active for several weeks outside mitochondria. This enabled us to study the structure of $F_1F_o$ ATP synthase in situ in a lipid bilayer. Sub-tomogram averaging of 2D crystals revealed the structure of bovine heart $F_1F_o$ ATP synthase in the membrane to approximately 24 Å. A model of molecular packing, determined by repositioning the sub-tomogram average into the tomographic volume, indicates that the membrane domain of the monomeric $F_1F_o$ ATP synthase from bovine heart bends the lipid bilayer.

### Map resolution

Our structure of the bovine heart $F_1F_o$ ATP synthase reconstituted into a lipid bilayer is consistent with the monomeric detergent-solubilised complex determined by single-particle analysis (*Baker et al., 2012*). Even though the resolution of our sub-tomogram average at 24 Å is nominally less good than that of the single-particle map at 18 Å, several features are more clearly resolved. This includes the densities of the central and peripheral stalks, the OSCP subunit and the α- and β-subunits, which are clearly separated in the sub-tomogram average (*Figure 4*). This may be a result of the more accurate particle alignment, which used 3D volumes rather than 2D projection images. In fact, *Baker et al. (2012)* recently reported that by reducing their data set by 82% to include only the best projection images, structural features of their cryo-EM map became clearer, although the nominal resolution, determined by the Fourier shell correlation, was unchanged. The resolution of our sub-tomogram average is about 1.5× better in the x–y plane than the z direction due to the single orientation of $F_1F_o$ ATP synthase particles in the crystal relative to the missing wedge of information (*Radermacher, 1988*).

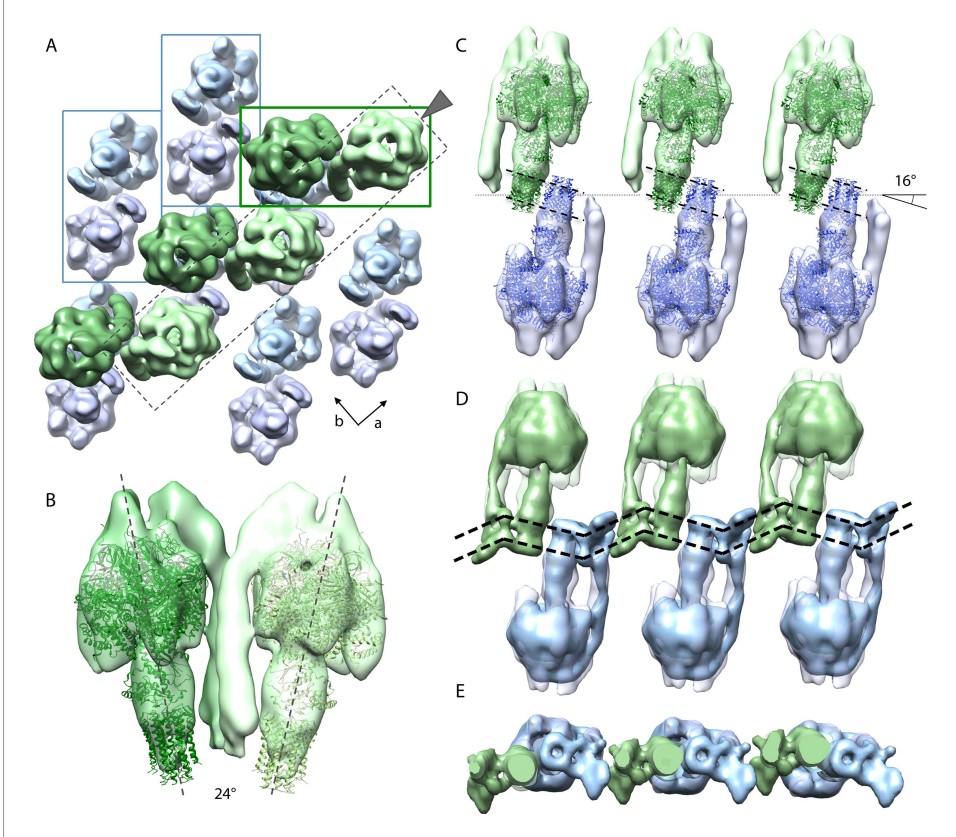

**Figure 5**. Packing of bovine $F_1F_o$ ATP synthase in the 2D crystal. (**A**) Top view of the 2D crystal lattice of bovine $F_1F_o$ ATP synthase. Rectangles indicate the position of ATP synthase pairs. Arrows indicate cell axes. (**B**) Side view of one pair. (**C**) Cross-section of the crystal lattice indicated by the dashed box and arrowhead in (**A**). $F_1F_o$ ATP synthases of opposite orientation in the membrane are connected via close interaction of their rotor-rings. The rotor-ring pairs are oriented 16° relative to the crystal plane (single grey dashed line). (**D**) Cross-section as in (**C**) but with the single-particle EM map (*Baker et al., 2012*) fitted into the subtomogram averages. The arrangement of the monomeric complexes on the 2D crystal lattice results in a locally kinked lipid bilayer (bold dashed lines). (**E**) Top view of the cross-section in (**D**) clipped to remove the catalytic domains of the upper $F_1F_o$ ATP synthase layer.

The following figure supplement is available for figure 5:

**Figure supplement 1**. Lattice disorder.

## The peripheral stalk

In our sub-tomogram average, the peripheral stalk is positioned along the non-catalytic αβ interface as previously reported (*Rees et al., 2009*; *Baker et al., 2012*) but is more offset towards the α-subunit than in the x-ray structure (*Rees et al., 2009*; *Figure 4*). A small connecting density visible between the midpoint of the peripheral stalk and α-subunit may account for this difference. A similar connection is seen in the single-particle cryo-EM map of the bovine complex (*Baker et al., 2012*). Analysis of the fitted x-ray structure suggests possible ionic interactions between subunit *d* and residues 463–475 of the α-subunit, which is the point where the structures of the α and β-subunits diverge (*Walker et al., 1982*). This additional contact may help the peripheral stalk in its role as a stator, but may also prevent the stalk from interfering with the catalytic cycle of the β-subunit by pulling it away from the α/β interface. Alternatively, the peripheral stalk may be pulled away from the α/β interface by its interactions in the membrane.

The dominant density of the peripheral stalk in the sub-tomogram average suggests that this region is more rigid than other parts of the complex. In accordance with this, we were able to fit the

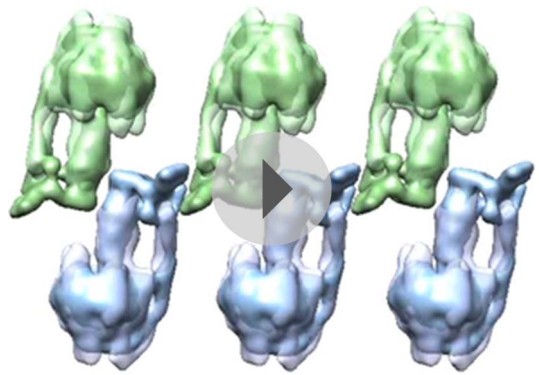

**Video 2.** Organization of bovine $F_1F_o$ ATP synthases in 2D crystals. The relative orientation of the $F_1F_o$ ATP synthases in a particularly well-ordered region of a 2D crystal is visualized by showing the re-inserted sub-tomogram averages, then by fitting the $F_1$-$c_8$ crystal structure and finally by fitting the single-particle map of *Baker et al., 2012*. $F_1F_o$ ATP synthases are orientated 16˚ to the crystal plane resulting in a zigzag arrangement of the lipid bilayer.

bovine heart peripheral stalk fragment (*Dickson et al., 2006*) extended by several residues from the $F_1$-peripheral stalk structure (*Rees et al., 2009*) into the density as a rigid body without the need to introduce hinges as previously suggested (*Baker et al., 2012*) (*Figure 4—figure supplement 1*). This fit therefore challenges the notion that the peripheral stalk acts as a flexible linker that stores torque or elastic energy during the catalytic cycle (*Sorgen et al., 1998*; *Sorgen et al., 1999*). This hypothesis was based on the bacterial enzyme, which has a peripheral stalk consisting of only two long alpha helices. The bovine peripheral stalk, in contrast, consists of one long alpha helix plus subunits $d$, $F_6$ and the N-terminal domains of other $F_o$ subunits, which would generate a more rigid structure (*Dickson et al., 2006*). A rigid peripheral stalk would hold the $(\alpha\beta)_3$ hexamer in a stationary position relative to the membrane. This is probably more important in mitochondria than in bacteria, as the mitochondrial $F_1F_o$ ATP synthase has another important role in generating local membrane curvature and maintaining cristae morphology through the formation of dimer rows (*Davies et al., 2012*). The dimer interface is located at the base of the peripheral stalk (*Davies et al., 2011*, *2012*) and thus any flexibility in the structure of the peripheral stalk may compromise the formation and stability of the dimer.

## The membrane region

Despite the prominent structural features visible in the membrane-extrinsic regions of the $F_1F_o$ ATP synthase, the membrane-embedded parts in our sub-tomogram average were not well-resolved (*Figure 3*). This was surprising as the $c$-ring, located directly beneath the central stalk, is easily visible in x-ray models filtered to 20 Å resolution. Therefore, if the sub-volumes are well enough aligned to give an overall resolution of 24 Å, we would expect to see the $c$-ring in the lipid bilayer. In our sub-tomogram average, we only see a weak density for the $c$-ring. The lack of detail in the membrane region is unlikely to be caused by the missing wedge as we have observed a similar phenomenon in sub-tomogram averages of other membrane proteins, which have no missing wedge of information, and in single-particle maps when the resolution does not extend beyond 15 Å. The extent of this problem appears to correlate with the composition of the surrounding lipid or detergent. Thus, the lack of protein density in the membrane region for our averages at this resolution seems to be caused mainly by contrast matching between the surrounding lipid and protein.

## 2D crystal processing of tomographic slices

The projection images generated from selected z-slices of tomographic volumes enabled us to use electron crystallographic image processing, even though the coherently packed areas of the 2D crystals were small. This improved the signal-to-noise ratio because noise originating from additional crystal layers or surrounding buffer was removed in silico by careful selection of the z-slices and Fourier filtering of the diffraction pattern. Thus 2D processing of tomographic 3D crystal volumes poses an attractive alternative for the electron crystallographic treatment of small 2D crystals of large membrane complexes that are not accessible to canonical image processing due to limited size and order. As exemplified in our 2D crystal of the $F_1F_o$ ATP synthase, the molecular packing of multi-subunit membrane complexes with large extramembranous domains can be complicated with multiple layers and multiple tilt angles relative to the crystal plane. These are circumstances under which projection maps alone are easily misinterpreted and our approach of combining electron

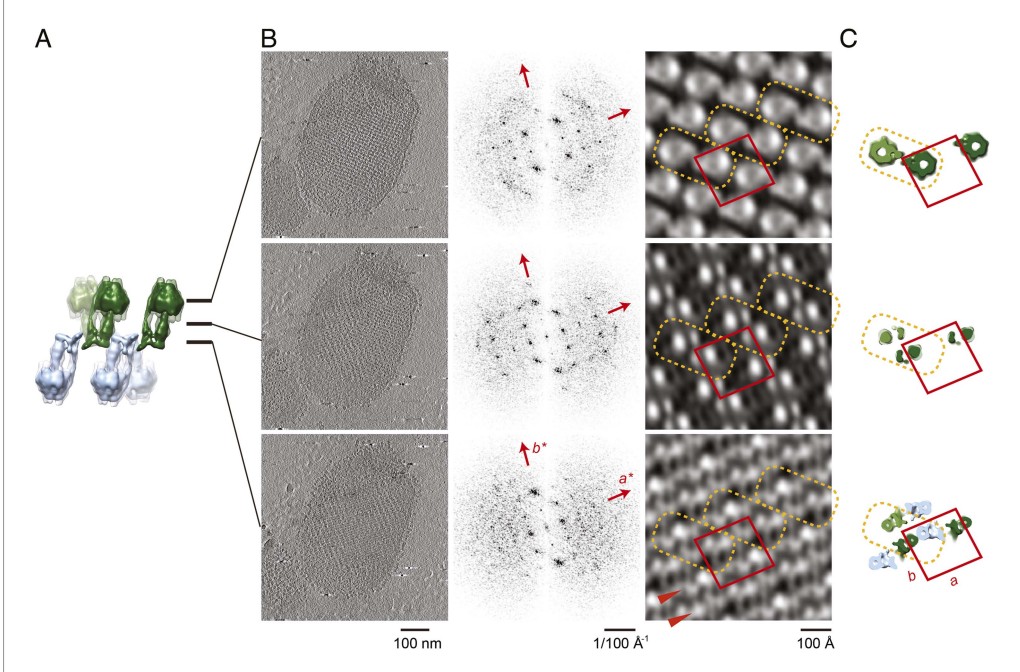

**Figure 6**. Projection maps of different z-slices in a 2D crystal of $F_1F_o$ ATP synthases. (**A**) Side view of crystal packing from *Figure 4*. (**B**) Projection images (left) calculated from z-slices of the tomographic volume at z-height positions indicated by black bars in (**A**), their Fourier transforms (centre) and projection maps (right). (**C**) z-slices through the crystal packing in (**A**) corresponding to the position of the projection maps shown in (**B**). (**A–C**) Protein densities observed in the projection maps perfectly match the features shown in the corresponding z-slices of the crystal-packing model. Dashed orange outlines indicate a pair of $F_1F_o$ ATP synthases, red boxes indicate the unit cell of the crystal with dimensions of a = 179.1 Å, b = 171.4 Å, γ = 94.9°. Red arrowheads in the lower panel of (**B**) indicate lines of continuous protein density in the membrane plane.

The following figure supplements are available for figure 6:

**Figure supplement 1**. Flow chart of electron crystallographic image processing of a tomographic volume.

**Figure supplement 2**. Unit cell parameters and crystal symmetry.

tomography with electron crystallography enables a straightforward interpretation (*Gerle et al., 2006*; *Tani et al., 2013*).

## $F_1F_o$ ATP synthase and membrane curvature

In mitochondria, the $F_1F_o$ ATP synthase forms rows of dimers along the highly curved ridges of cristae membranes (*Strauss et al., 2008*; *Davies et al., 2011*). Disruption of dimers prevents the formation of wild-type cristae and increases the generation time of the organism (*Paumard et al., 2002*; *Davies et al., 2012*). Molecular dynamics simulations have shown that the $F_1F_o$ ATP synthase dimer structure alone is sufficient to bend the lipid bilayer and drive row formation (*Davies et al., 2012*). In addition, single-particle cryo-EM of mitochondrial ATP synthase complexes indicates that the bend of the dimer may originate from the structure of the $F_o$ domain in a single monomer (*Baker et al., 2012*).

Through our analysis of the $F_1F_o$ ATP synthase packing in 2D crystals, we have shown that the $F_o$ domain of monomeric bovine heart $F_1F_o$ ATP synthase indeed induces a substantial kink in the lipid bilayer (*Figures 5, 6*) explaining in molecular terms the zigzag appearance of the membrane in the 2D crystals (*Figure 2*). When two monomeric complexes join to form a dimer in the inner mitochondrial membrane, the resulting angle between the long axes of the monomers would be ~86° (*Figure 7*) exactly as observed in the subtomogram averages of the bovine $F_1F_o$ ATP synthase dimer

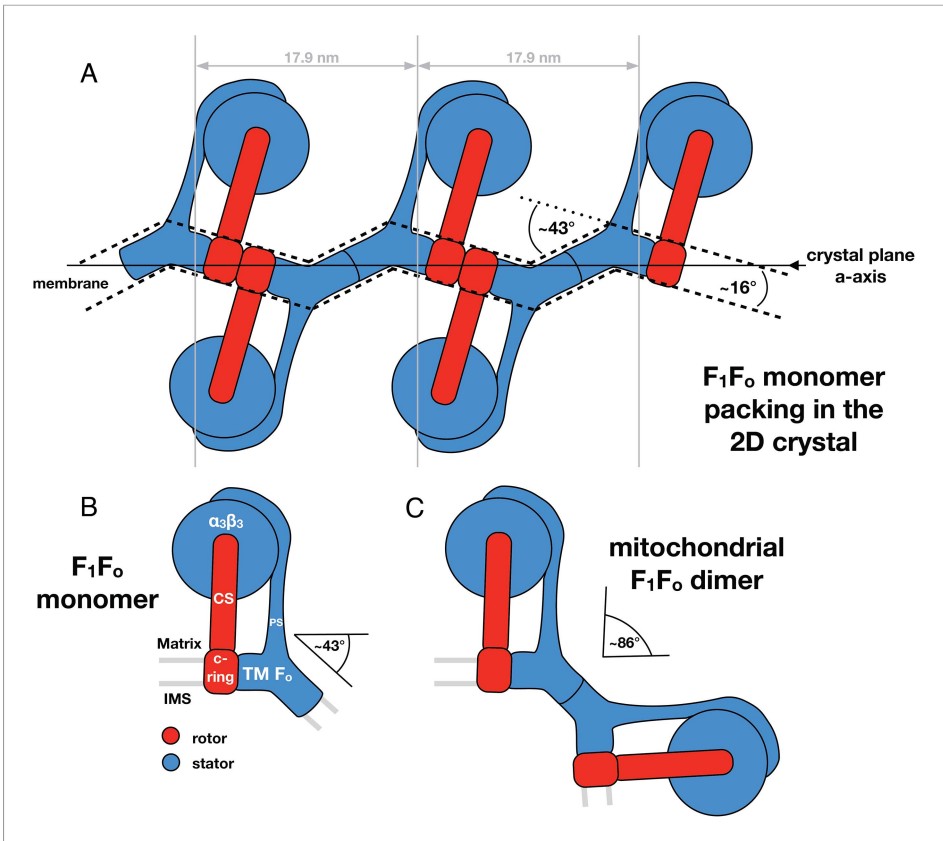

**Figure 7**. Monomeric mitochondrial $F_1F_o$ ATP synthase as the factor determining cristae membrane curvature. (**A**) Schematic diagram of bovine $F_1F_o$ ATP synthase in the 2D crystal lattice, with its transmembrane $F_o$ stator domain imposing a local 43° kink and a 16° inclination of the *c*-ring relative to the crystal plane. Blue, stator; red, rotor. (**B**) Schematic diagram of a single monomeric bovine $F_1F_o$ ATP synthase. CS: central stalk; PS: peripheral stalk; IMS: intermembrane space; TM $F_o$: transmembrane domain of $F_o$. (**C**) Association of two monomeric $F_1F_o$ complexes into a dimer results in an angle of 86°, as observed in ATP synthase dimers in mitochondrial cristae (***Davies et al., 2012***).

(***Davies et al., 2012***). This dimer angle gives rise to the high local curvature of membranes required to shape the cristae (***Davies et al., 2012***). We conclude that the $F_1F_o$ monomer is the basic building block of the ATP synthase dimer rows. Newly assembled monomeric complexes of the bovine heart $F_1F_o$ ATP synthase monomers would converge on pre-existing regions of high membrane curvature where they assemble into dimers, thus propagating the self-association of dimers into rows.

## Perspective

Using a combination of electron cryo-tomography, sub-tomogram averaging, and electron crystallography, we have determined the structure of an $F_1F_o$ ATP synthase in a lipid bilayer. This in vitro system is free from additional proteins, the enzyme is fully functional, and the proteoliposomes are in principle proton-tight. Therefore, this system may be suitable for studying the effect of physiological parameters such as ΔpH, Δpmf, and substrate availability on the structure and function of $F_1F_o$ ATP synthase (***Walker, 2013***). In addition, with the recent advancements in electron detector development and image processing methods in cryo-EM, it may soon be possible to generate tomographic volumes with sub-nanometer resolution routinely (***Schur et al., 2013***). This resolution should be sufficient to determine whether the structure of the enzyme changes under different physiological conditions. Only when these questions are answered will we really understand how the $F_1F_o$ ATP synthase works in the membrane.

## Materials and methods

### Purification of intact $F_1F_o$ ATP synthase from bovine heart mitochondria

$F_1F_o$ ATP synthase was purified by a published procedure (*Maeda et al., 2013*). Briefly, sub-mitochondrial particles in 40 mM HEPES pH 7.8, 2 mM $MgCl_2$, 0.1 mM EDTA, and 0.1 mM DTT, isolated from fresh bovine hearts as described previously (*Shinzawa-Itoh et al., 2010*), were solubilized on ice by the addition of deoxycholate and decylmaltoside to final concentrations of 0.7% (wt/vol) and 0.4% (wt/vol), respectively. Subsequently, the suspension was centrifuged at 176,000×*g* for 50 min and the supernatant applied to a sucrose step gradient (40 mM HEPES pH 7.8, 0.1 mM EDTA, 0.1 mM DTT, 0.2% [wt/vol] decylmaltoside and 2.0 M, 1.1 M, 1.0 M, or 0.9 M sucrose) and centrifuged overnight at 176,000×*g* for 15.5 hr. Fractions with ATPase activity were loaded onto a Poros-20HQ ion-exchange column and eluted by a linear concentration gradient of 0–240 mM KCl in 40 mM HEPES pH 7.8, 2 mM $MgCl_2$, 0.1 mM EDTA, 0.1 mM DTT and 0.2% (wt/vol) decylmaltoside. $F_1F_o$ ATP synthase fractions containing high amounts of native phospholipids were concentrated to 10 mg/ml (Advantec Ultra filter, polysulfone, MWCO 200 kDa, Toyo Roshi, Tokyo, Japan).

### Membrane reconstitution and two-dimensional crystallisation

Synthetic, chemically pure 1,2-dimyristoyl-sn-glycero-3-phosphocholine (Avanti Polar Lipids, Alabaster, AL) was solubilised in decylmaltoside and mixed with freshly purified bovine $F_1F_o$ ATP synthase at a lipid to protein ratio of 0.2. After overnight incubation on ice, the detergent was removed by dialysis using 20 µl Hampton dialysis buttons (Hampton Research, Aliso Viejo, CA) (membrane cutoff 15,000 Da, SpectraPor#7, Spectrum, Los Angeles, CA) and 500 ml of dialysis buffer (40 mM Tris-HCl pH 8.2, 100 mM NaCl, 0.02% [wt/vol] $NaN_3$, 0.5 mM ADP, 5 mM $MgCl_2$, 0.1 mM DTT, 0.1 mM EDTA). The dialysis buffer was exchanged daily and the sample was incubated at 27°C for 10 to 21 days.

### ATPase activity measurement, gel electrophoresis, and mass spectrometry

To determine the specific enzymatic activity and the proportion of coupled complexes of both the isolated $F_1F_o$ ATP synthases and the crystalline vesicles resolubilized with 4% (wt/vol) digitonin, an ATP-regenerating enzyme-coupled assay was used (*Pullman et al., 1960*). The hydrolysis of ATP by the $F_1F_o$ ATP synthase was followed by NADH oxidation at 340 nm at 20°C in the absence or presence of oligomycin. To confirm the subunit composition and intactness of the bovine $F_1F_o$ ATP synthase, crystalline vesicles, resolubilized with sodium dodecyl sulfate were examined by denaturing SDS-PAGE and vesicles resolubilized with 4% (wt/vol) digitonin by non-denaturing blue-native PAGE (*Wittig et al., 2006*). The presence of the lower molecular weight subunits (5000–12,000 Da) of the $F_1F_o$ ATP synthase in the crystalline vesicles was confirmed by MALDI-TOF mass spectrometry (Bruker Daltonics Inc., MA, USA).

### Negatively stained electron microscopy

2.5 µl of dialysed sample was applied to freshly glow-discharged, carbon-coated 400 mesh copper grids (Veco, Nisshin, Tokyo, Japan), blotted and stained with 2% uranyl acetate solution. Grids were screened using a JEM1010 transmission electron microscope (JEOL, Tokyo, Japan) at 100 kV and images were acquired using a 2k × 2k slow-scan CCD camera (Gatan, Pleasanton, CA). Images were recorded at a magnification of 40,000×, which corresponds to a pixel size of 6 Å using a 2 s exposure.

### Electron cryo-tomography

Samples of 2D crystals were screened by negative stain EM immediately before freezing. Samples containing well-ordered arrays of $F_1$ heads were selected and mixed 1:1 with fiducial gold markers (6 nm gold particles conjugated to protein A, Aurion). 3 µl of the protein:gold sample were applied to glow-discharged quantifoil grids (R2/2, 300 copper mesh), blotted for 3 s (#4 Whatman paper, Sigma-Aldrich, St. Louis, MI) and plunge-frozen in liquid ethane using a home-built freezing device. Single-axis tilt series (±60°, step size 1.5°) were collected on an FEI Krios microscope (FEI, Hillsboro, OR) operating at 300 kV and equipped with a post-column energy filter (GIF Quantum, Gatan) with a K2 summit direct detector in counting mode (Gatan). Images were taken with a specimen pixel size of

0.33 nm and a defocus of 2.5 μm. Tilt series were aligned using the gold fiducials and back-projected to generate tomographic volumes using the IMOD package (*Kremer et al., 1996*).

## Sub-tomogram averaging

For particle picking, tomograms were binned 4 × 4 and filtered by nonlinear anisotropic diffusion to increase contrast (*Frangakis and Hegerl, 2001*). $F_1$ subcomplexes were picked manually in 3dmod with the $F_1$ subcomplexes of opposite orientation assigned to different files. Particles were separated randomly into two data sets and processed independently. Particle alignment was performed in PEET, starting with the contrast-enhanced binned 2 × 2 tomograms, then the binned 2 × 2 tomograms without contrast enhancement and finally the unbinned (1 × 1) volume. An initial reference volume was calculated for each data set by averaging all selected subvolumes on one side of the membrane. Subvolumes containing $F_1$ subcomplexes from the opposite side of the membrane were rotated 180° about the x-axis prior to alignment. Alignment parameters were then restricted to ±45° about the x and y-axis, and 12 pixels in xyz. No restriction was placed on the z-axis rotation. Initial alignment was performed with a resolution limit of 60 Å and gradually lowered to 30 Å in accordance with the FSC. To assess over-fitting, phases beyond 40 Å in the tomogram were randomized and the last alignment iteration was repeated (*Chen et al., 2013*). The final average, calculated from 2500 subvolumes, was filtered to 18 Å using a Fermi filter with a temperature factor of 0.002 $px^{-1}$ (*Frank, 1996*). Fourier shell correlation was performed in PEET using a box size of 96 voxels. Atomic models were docked into the EM density using the sequential fit routine of Chimera and superimposed using the matchmaker command (*Pettersen et al., 2004*). Electron density maps of atomic models were calculated in CCP4 (*Winn et al., 2011*).

## Image analysis of tomographic slices

Two types of projection images were prepared from tomographic volumes of a single vesicular 2D crystal (for a flow chart see *Figure 6—figure supplement 1*). To determine the lattice parameters, a projection image of a single-layered 2D crystal was calculated from the corresponding z-slices of the tomographic volume by first extraction and conversion into a projection image using the IMOD package (*Kremer et al., 1996*) (*Figure 6—figure supplement 2A*—left panel) and then processing the projection images using the MRC image processing programmes (*Crowther et al., 1996*) (*Figure 6—figure supplement 2A*, middle and right panel). The unit cell parameters of the crystal were: a = 179.1 Å, b = 171.4 Å, γ = 94.9°. As the programme ALLSPACE (*Valpuesta et al., 1994*) indicated only one plane of symmetry, data were merged in *p*1 (*Figure 6—figure supplement 2B*). The single-layered 2D crystal in the tomographic volume was subdivided along the z-axis and a total of 64 projection images were generated at an interval spacing of 6.66 Å, corresponding to the sampling size of the tomographic volume. All images were processed with the MRC image processing programmes to correct for crystal lattice distortion (*Henderson et al., 1986*). Only a few reflections had statistically significant amplitudes beyond 30 Å resolution and all projection maps were calculated within a 30 Å resolution limit.

## Acknowledgements

We would like to like to thank Yoshito Kaziro for help and advice during the planning stages of the project and Shinya Yoshikawa for encouragement and support. We gratefully acknowledge Kazumi Kobayashi for maintenance of the JEOL electron microscopes and Tomoko Maeda for assistance with $F_1F_o$ ATP synthase purification and John Heumann for writing the randomised phase program for tomographic data. This work was supported by the Max Planck Society (KMD, DJM, and WK), the Cluster of Excellence Frankfurt 'Macromolecular Complexes' supported by the DFG (KMD and WK), the Special Coordination Fund for Promoting Science and Technology of MEXT, Japan (to CG), a Platform for Drug Design, Discovery and Development grant from MEXT, Japan (to CG), the JST/CREST (to TT and CG), a Grants-in-Aid for Scientific Research (S) from MEXT, Japan, the New Energy and Industrial Technology Development Organization (NEDO) and the National Institute of Biomedical Innovation (to YF) and a Grants-in-Aid for Scientific Research (C) from MEXT, Japan (to KT). Establishment, continuous execution and improvement of bovine $F_1F_o$ ATP synthase purification was supported by a Grants-in-Aid for Scientific Research (A) (to SY and KS) from MEXT, Japan, a Grants-in-Aid for the Global Center of Excellence Program

(to SY and KS) from MEXT, Japan and a Targeted Protein Research Program (to SY and KS) from MEXT, Japan.

## Additional information

### Competing interests

WK: Reviewing editor, *eLife*. The other authors declare that no competing interests exist.

### Funding

| Funder | Grant reference | Author |
|---|---|---|
| Max-Planck-Gesellschaft | | Karen M Davies, Deryck J Mills, Werner Kühlbrandt |
| Deutsche Forschungsgemeinschaft | Cluster of Excellence Frankfurt "Macromolecular Complexes" | Karen M Davies, Werner Kühlbrandt |
| Ministry of Education, Culture, Sports, Science, and Technology (MEXT) | Special Coordination Fund | Christoph Gerle |
| Ministry of Education, Culture, Sports, Science, and Technology (MEXT) | Platform for Drug Design, Discovery and Development grant | Christoph Gerle |
| New Energy and Industrial Technology Development Organisation (NEDO) | | Yoshinori Fujiyoshi |
| National Institute of Biomedical Innovation (NIBIO) | | Yoshinori Fujiyoshi |
| Core Research for Evolutional Science and Technology, Japan Science and Technology Agency (CREST, JST) | | Tomitake Tsukihara, Christoph Gerle |
| Ministry of Education, Culture, Sports, Science, and Technology (MEXT) | Grants-in-Aid for Scientific Research | Yoshinori Fujiyoshi, Kazutoshi Tani, Kyoko Shinzawa-Itoh |
| Ministry of Education, Culture, Sports, Science, and Technology (MEXT) | Grants-in-Aid for the Global Center of Excellence Program | Kyoko Shinzawa-Itoh |
| Ministry of Education, Culture, Sports, Science, and Technology (MEXT) | Targeted Protein Research Program | Kyoko Shinzawa-Itoh |

The funders had no role in study design, data collection and interpretation, or the decision to submit the work for publication.

### Author contributions

CJ, Performed protein purification, Biochemical assays, 2D crystallization and negative stain EM; KMD, Initiated the project, Designed research, Performed cryo-EM, Tomographic data analysis and modelling, Analyzed the data, Interpreted the results, Wrote the manuscript; KS-I, Established and performed protein purification and biochemical assays; KT, Performed electron crystallographic image analysis, Discussed results; SM, Performed protein purification and biochemical assays; DJM, Assisted cryo-EM data accquisition; TT, Contributed key reagents and discussed the results; YF, Guided the project, Discussed results, Contributed key reagents; WK, Initiated the project, Designed research, Analysed the data, Interpreted the results, Wrote the manuscript; CG, Conceived and initiated the project and designed research, Analyzed the data, Interpreted the results, Wrote the manuscript

### Author ORCIDs

Christoph Gerle, http://orcid.org/0000-0002-7265-2804

## Additional files

### Major datasets

The following datasets were generated:

| Author(s) | Year | Dataset title | Dataset ID and/or URL | Database, license, and accessibility information |
|---|---|---|---|---|
| Jiko C, Davies KM, Shinzawa-Itoh K, Tani K, Maeda S, Mills DJ, Tsukihara T, Fujiyoshi Y, Kühlbrandt W, Gerle C | 2015 | Sub-tomogram average of a mammalian F-type ATP synthase monomer | http://www.ebi.ac.uk/pdbe/entry/emdb/EMD-2982 | Publicly available at the Electron Microscopy Data Bank (EMDB) at PDBE. |
| Jiko C, Davies KM, Shinzawa-Itoh K, Tani K, Maeda S, Mills DJ, Tsukihara T, Fujiyoshi Y, Kühlbrandt W, Gerle C | 2015 | Raw images of two tilt-series obtained from mammalian $F_1F_o$ 2D crystals that were used for tomographic reconstructions | http://www.ebi.ac.uk/pdbe/emdb/empiar/entry/10027I | Publicly available at the Electron Microscopy Data Bank (EMDB) at PDBE. |

The following previously published datasets were used:

| Author(s) | Year | Dataset title | Dataset ID and/or URL | Database, license, and accessibility information |
|---|---|---|---|---|
| Rees DM, Leslie AGW, Walker JE | 2009 | The structure of the membrane extrinsic region of bovine ATP synthase | http://www.rcsb.org/pdb/explore/explore.do?structureId=2WSS | Publicly available at RCSB Protein Data Bank 2WSS. |
| Kane Dickson V, Silvester JA, Fearnley IM, Leslie AGW, Walker JE | 2006 | Subcomplex of the stator of bovine mitochondrial ATP synthase | http://www.rcsb.org/pdb/explore/explore.do?structureId=2CLY | Publicly available at RCSB Protein Data Bank 2CLY. |
| Watt IN, Montgomery MG, Runswick MJ, Leslie AGW, Walker JE | 2010 | Crystal structure of bovine F1-C8 sub-complex of ATP synthase | http://www.rcsb.org/pdb/explore/explore.do?structureId=2XND | Publicly available at RCSB Protein Data Bank 2XND. |
| Abrahams JP, Leslie AG, Lutter R, Walker JE | 1994 | Bovine mitochondrial F1-ATPase | http://www.rcsb.org/pdb/explore/explore.do?structureId=1BMF | Publicly available at RCSB Protein Data Bank 1BMF. |

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
