## [Decision Letter]

Thank you for sending your work entitled “Bovine F_1_F_o_ ATP synthase monomers bend the lipid bilayer in 2D membrane crystals” for consideration at *eLife*. Your article has been evaluated by John Kuriyan (Senior editor), a Reviewing editor, and four reviewers, and we invite you to submit a revision.

The following individuals responsible for the peer review of your submission have agreed to reveal their identity: Sjors Scheres (Reviewing editor) and Henning Stahlberg (peer reviewer). Three other reviewers remain anonymous.

The Reviewing editor and the reviewers discussed their comments before we reached this decision, and the Reviewing editor has assembled the following comments to help you prepare a revised submission.

All reviewers much appreciated the fact that 2D crystals of active and intact complexes were obtained, as well as the novel image processing procedures that were used to analyse the tomographic data of these crystals. All reviewers also agreed that the main conclusion of the paper, that the membrane part of the F_1_F_o_ ATP synthase monomer by itself bends the lipid bilayer, is very interesting and supportive of earlier results. However, a majority of the reviewers did feel that this conclusion needs additional experimental evidence. In particular the fact that the zig-zag pattern of the monomers is only inferred from the crystal packing, and not observed directly, was judged to be an important weakness of the current manuscript. This should be adequately addressed in a revised version. Possible options discussed were the inclusion of a cross-section plot from the edges of a vesicular 2D crystal, the inclusion of subtomograms from such edges in the averaging procedure to decrease the effects of the missing wedge on the membrane region in the subtomogram average, recording dual-axis tomograms to 70°, or recording images on cryo-sections of crystals.

In addition, two reviewers raised concerns about the possibility to distinguish between different conformations of the alpha/beta pairs at this resolution. Such claims should be backed by more information about the fittings of the different conformations in the maps. In addition, to exclude that random noise in the map could lead to one conformation fitting better than another, the authors could refine two halves of the data independently (i.e. gold-standard procedure, so a FSC=0.143 criterion could be used to assess resolution) and perform the fittings of the different conformations in both half-maps.

Lastly, it was felt that Figure 6 is not representative of the actual results in the paper. It should contain a panel with the relative orientations of the monomers in the 2D crystal, and how the 20° inclination of the c-ring, together with the 2D crystal lattice parameters, lead to the same 43° angle as observed in the single-particle reconstruction (which also isn't very clear from the paper). This figure should also show that the crystal packing is actually very different from the 86° dimer.

Minor comments:

How was the resolution of the tomography reconstruction of 21Å determined? The authors list FSC 0.5 as criterion. But was any Gold Standard method applied here? What about reference bias?

Why is the unit cell given as p1 symmetry? This should be p121b or something similar. Or do the authors merely state that they didn't apply any symmetrization? If so, then they should state it as such.

Figure 1—figure supplement 3 should have reference markers.

It would be helpful to have an introductory sentence in the Results section describing the crystalline vesicles so that the reader can understand that '2D crystals', 'crystalline tubes', 'rectangular-shaped vesicles', 'tubular 2D crystals', and 'tubular crystals' all mean the same thing.

Figure 1: the fact that there are many more ATPase heads on the insides of the vesicles than on the outsides (whereas there should be equal numbers) suggests there may be some degradation, despite the biochemical analyses suggesting the enzyme is complete and active?

I cannot understand Figure 1—figure supplement 3. Why do the gels show mixtures of monomers and dimers, and why do the digitonin-resolubilised 2D crystals have a larger proportion of dimers? I don't understand the significance of the two curves in B. They need to be explained more fully in the caption.

The discussion on the membrane region (first paragraph of the subsection headed “The membrane region”) is not good. The lack of detail in the membrane region is surely due simply to poor resolution perpendicular to the membrane associated with the (quite substantial) missing wedge. It is not a contrast-matching issue. Analyses of helical tubes, where there is no such missing information, have demonstrated clearly that densities of a protein are well contrasted against the lipid bilayer at resolutions near 20Å, and certainly do not require a resolution of 8Å to be visible.

---

## [Author Response]

*All reviewers much appreciated the fact that 2D crystals of active and intact complexes were obtained, as well as the novel image processing procedures that were used to analyze the tomographic data of these crystals. All reviewers also agreed that the main conclusion of the paper, that the membrane part of the F*_*1*_*F*_*o*_
*ATP synthase monomer by itself bends the lipid bilayer, is very interesting and supportive of earlier results. However, a majority of the reviewers did feel that this conclusion needs additional experimental evidence. In particular the fact that the zig-zag pattern of the monomers is only inferred from the crystal packing, and not observed directly, was judged to be an important weakness of the current manuscript. This should be adequately addressed in a revised version. Possible options discussed were the inclusion of a cross-section plot from the edges of a vesicular 2D crystal, the inclusion of subtomograms from such edges in the averaging procedure to decrease the effects of the missing wedge on the membrane region in the subtomogram average, recording dual-axis tomograms to 70°, or recording images on cryo-sections of crystals*.

Thank you for this suggestion. We have now visualized the zigzag membrane morphology directly by collecting tomograms of crystalline vesicles in thick vitreous ice (>200nm). One of these vesicles was optimally orientated relative to the electron beam to reveal the zigzag membrane structure both in projection and in cross-sections of the tomographic volume. These images have been compiled into a new figure, which is now Figure 2.

*In addition, two reviewers raised concerns about the possibility to distinguish between different conformations of the alpha/beta pairs at this resolution. Such claims should be backed by more information about the fittings of the different conformations in the maps. In addition, to exclude that random noise in the map could lead to one conformation fitting better than another, the authors could refine two halves of the data independently (i.e. gold-standard procedure, so a FSC=0.143 criterion could be used to assess resolution) and perform the fittings of the different conformations in both half-maps*.

We have recalculated the FSC using gold standard and randomized phase procedures. In addition, we have made a direct comparison of our subtomogram average to a low-resolution map generated by Fourier filtering the x-ray structure of the F_1_-subcomplex to 25 Å. The different positions of the alpha/beta subunits in the hexamer are clearly visible at this resolution (Figure 3—figure supplement 2) but not their different conformations. We have thus deleted the sentence “the conformation of each /β pair varies within the hexamer, suggesting different functional states” from the manuscript.

*Lastly, it was felt that*
Figure 6
*is not representative of the actual results in the paper. It should contain a panel with the relative orientations of the monomers in the 2D crystal, and how the 20° inclination of the c-ring, together with the 2D crystal lattice parameters, lead to the same 43° angle as observed in the single-particle reconstruction (which also isn't very clear from the paper). This figure should also show that the crystal packing is actually very different from the 86° dimer*.

As requested, we have added a new panel (Figure 7) to the previous Figure 6 (now Figure 7) to show the orientation of the ATP synthase monomers in the 2D crystals relative to the crystal lattice parameters. This panel clearly illustrates the ∼20° (actually 16°, the more accurate angle now used throughout the paper) inclination of the c-ring relative to the 2D crystal plane which results in the membrane forming two 43° bends in order to embedded neighbouring invert c-ring pairs in the bilayer.

Panels B and C of Figure 7 (previously panels A and B of Figure 6) illustrate how the angles observed in the 2D crystal relates to the ATP synthase monomer and dimer.

*Minor comments*:

How was the resolution of the tomography reconstruction of 21Å determined? The authors list FSC 0.5 as criterion. But was any Gold Standard method applied here? What about reference bias?

The resolution of 21Å was calculated by splitting the dataset in two after alignment. There was no reference bias as the initial model was calculated ab initio from the subvolumes.

We have now repeated the subtomogram averaging using gold standard procedures and included a gold standard FSC analysis on averages calculated from tomograms with phases randomized beyond 40 Å resolution (Figure 3—figure supplement 1). We have also expanded the description of the subtomogram averaging procedure in the Methods section.

*Why is the unit cell given as p1 symmetry? This should be p121b or something similar. Or do the authors merely state that they didn't apply any symmetrization? If so, then they should state it as such*.

The crystals are poorly ordered, and the programme ALLSPACE did not detect a clear, plane group symmetry (Figure 6—figure supplement 1). Thus all 2D processing was performed without symmetry applied.

Figure 1—figure supplement 3
*should have reference markers.*

The point of BN-PAGE shown in Figure 1—figure supplement 3 was to demonstrate that protein reconstituted into vesicles and stored at 27°C for three weeks was of equal quality to freshly purified protein, in particular absence of degradation, disassembly or aggregation. Thus the migration profile of proteins solubilized from three-week-old 2D crystals using digitonin (lane 3 and 4) was directly compared to freshly purified bovine F_1_F_o_ ATP synthases (lane 1 and 2). The aim of this gel was simply to show that both samples behave in the same way, which they do, and not to determine the molecular mass of the ATP synthase monomer and dimer. These molecular masses are well known, so the bands act as their own markers, far better than any extrinsic marker of soluble proteins would do for a blue native PAGE of a membrane protein. The migration behavior of membrane protein complexes on such a gel is highly dependent not only on the size and charge of the protein but also the shape and amount of bound lipids or detergents.

*It would be helpful to have an introductory sentence in the Results section describing the crystalline vesicles so that the reader can understand that '2D crystals', 'crystalline tubes', 'rectangular-shaped vesicles', 'tubular 2D crystals', and 'tubular crystals' all mean the same thing*.

We have simplified the terminology and now just use the terms: crystalline vesicles and rectangular crystalline vesicles. The latter contains 2D crystals, which showed the highest degree of crystalline order.

Figure 1: *the fact that there are many more ATPase heads on the insides of the vesicles than on the outsides (whereas there should be equal numbers) suggests there may be some degradation, despite the biochemical analyses suggesting the enzyme is complete and active?*

We can exclude proteolytic degradation as a cause of missing ATPase heads, as the blue-native gel in Figure 1—figure supplement 3 shows clearly that this is not the case. We agree that Figure 1 in the original manuscript did seem to suggest that there were more ATPase heads on the inside. However, this figure was not representative, as the apparent number of ATPase heads on either side of the membrane depends on the exact position of the tomographic slice relative to the 2D lattice and the curvature of the membrane at the edge of the vesicle. In the revised manuscript, we have replaced Figure 1 by a new figure, in which the plane of the slice was chosen more carefully to indicate roughly similar numbers of heads on either side of the membrane. This is also evident from Figure 1 and from the new Figure 2 in the revised manuscript.

*I cannot understand*
Figure 1—figure supplement 3*. Why do the gels show mixtures of monomers and dimers, and why do the digitonin-resolubilised 2D crystals have a larger proportion of dimers? I don't understand the significance of the two curves in B. They need to be explained more fully in the caption*.

The F_1_F_o_ ATP synthase form rows of dimers in mitochondrial cristae (7). To isolate the F_1_F_o_ ATP synthase, these membranes were solubilized using decyl maltoside (DM). Although this detergent is relatively harsh, the purification procedure used in this study was mild enough to retain a significant portion of F_1_F_o_ ATP synthase dimers. Therefore a clear dimer band is seen in lane 1 and 2, as expected.

The 2D crystals were solubilized using the very mild detergent digitonin. It is well known that this detergent preserves oligomeric assemblies of mitochondrial F_1_F_o_ ATP synthase dimers upon membrane solubilisation. In the 2D crystal, the F_1_F_o_ ATP synthase interacts strongly via its c-ring with complexes of opposite orientation and weakly with complexes of the same orientation via its peripheral stalk. Some of these interactions are likely to be maintained during the solubilisation of the 2D crystals with the mild detergent digitonin.

As digitonin is a milder detergent than DM, more protein-protein interactions are preserved when the 2D crystals are solubilized with digitonin than when the mitochondrial membranes are solubilized with DM. Hence the larger proportion of dimers in lanes 3 and 4 compared to lanes 1 and 2.

To explain the two curves in B more clearly, we added the following statement to the caption:

“Hydrolysis of ATP by the F_1_F_o_ ATP synthase was monitored using an enzyme couple assay by detecting NADH oxidation at 340 nm at 20°C in the absence (blue trace) or presence (green trace) of oligomycin. Oligomycin inhibits ATP hydrolysis by binding to and stopping rotation of the F_o_ motor, i.e. oligomycin sensitivity is a measure of the relative proportion of coupled, intact F_1_F_o_ ATP synthase complexes in the digitonin resolubilised 2D crystals, in this case > 95%.”

*The discussion on the membrane region (first paragraph of the subsection headed “The membrane region”) is not good. The lack of detail in the membrane region is surely due simply to poor resolution perpendicular to the membrane associated with the (quite substantial) missing wedge. It is not a contrast-matching issue. Analyses of helical tubes, where there is no such missing information, have demonstrated clearly that densities of a protein are well contrasted against the lipid bilayer at resolutions near 20Å, and certainly do not require a resolution of 8Å to be visible*.

In our crystallographic projection map Fourier filtering allowed the visualization of transmembrane protein density in the presence of a dense and intact lipid bilayer even at 30 Angstrom resolution and a strong missing wedge. This is because the lipid bilayer has no regular order, and the Fourier terms originating from the membrane are filtered out during the masking of the protein diffraction pattern. A similar process also occurs during helical reconstructions, which means the signal from the protein is enhanced whereas the signal from disordered lipid is removed. In contrast, subtomogram averaging is conducted using real space cross correlation-based methods and all Fourier terms including those from the membrane contribute to the final average. This means both contrast matching and the missing wedge greatly affects the final average and both are likely to be important causes for the absence of transmembrane density in our subtomogram average.

The lack of protein density in the membrane region of our subtomogram average is not a consequence of the missing wedge. We have calculated subtomogram averages of several membrane-embedded ATP synthases that were randomly orientated in the tomographic volume. We found that every single average lacks protein density in the membrane region. We know from the single-particle averages of ATP synthase dimers (Allegretti et al., Nature 2015) published by us (KMD, WK) since the present manuscript was submitted that the c-ring density is seen clearly only at 10Å resolution or better, where membrane-spanning helices begin to become visible. At lower resolution, the mean scattering densities of the protein and surrounding lipid or detergent are too similar to show a clear boundary.